# Towards large language model-based personal agents in the enterprise: Current trends and open problems

**Vinod Muthusamy**
IBM Research

**Yara Rizk**
IBM Research

**Kiran Kate**
IBM Research

**Praveen Venkateswaran**
IBM Research

**Vatche Isahagian**
IBM Research

**Ashu Gulati**
Persistent Systems

**Parijat Dube**
IBM Research

## Abstract

There is an emerging trend to use large language models (LLMs) to reason about complex goals and orchestrate a set of pluggable tools or APIs to accomplish a goal. This functionality could, among other use cases, be used to build personal assistants for knowledge workers. While there are impressive demos of LLMs being used as autonomous agents or for tool composition, these solutions are not ready for mission-critical enterprise settings. For example, they are brittle to input changes, and can get stuck in reasoning loops. These use cases raise challenging problems opening up exciting areas of NLP research, such as trust and explainability, consistency and reproducibility, adherence to guardrails and policies, best practices for composable tool design, and the need for new metrics and benchmarks. This vision paper illustrates some examples of LLM-based autonomous agents that reason and compose tools, highlights cases where they fail, surveys some of the recent efforts in this space, and lays out the research challenges to make these solutions viable for enterprises.

## 1 Introduction

The emergence of ChatGPT has put large language models (LLMs) (Bommasani et al., 2021) under the microscope to determine what they can and cannot do (Hu, 2023). With LLMs outperforming state-of-the-art approaches for traditional areas of natural language processing (NLP) research (Yang et al., 2023a), the focus shifted to other ways that LLMs can be applied. The last few months, for example, have brought on a flurry of activity around LLMs as an orchestrator or brain of autonomous agents tasked with completing high level goals (Park et al., 2023; Wu et al., 2023a; Vemprala et al., 2023; Singh et al., 2022).

Autonomous agents perceive the world they operate in, reason about the events within it and their goal, decompose this goal into sub-problems, and

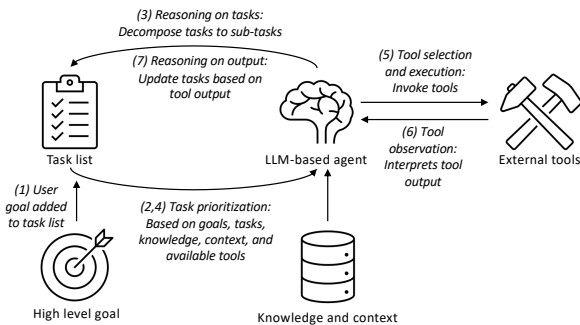

Figure 1: An LLM-based agent autonomously reasons about tasks and composes external tools to complete tasks, and ultimately achieve the user's goal.

perform a sequence of actions (equivalent to composing external tools and APIs to carry out tasks) to achieve a new state in this world (Russell, 2010). Figure 1 depicts the iterative pattern that some LLM-based autonomous agents are applying (Yao et al., 2022; Nakajima, 2023; Wang et al., 2023b; Brohan et al., 2023).

Philosophically, it is perhaps not surprising that language models are being used for reasoning and carrying out complex goals. For example, the Sapir-Whorf Hypothesis (Kay and Kempton, 1984) suggests that language influences thought. Better language models then should be capable of more sophisticated thinking, including reasoning and planning. Another connection is to the modes of thinking in humans: System 1 for instinctive thought processes, and System 2 for more deliberative thought (Kahneman, 2011). One can consider NLP tasks such as entity recognition as belonging to the System 1 bucket, whereas tasks where LLMs are used to reason and plan belong to the System 2 bucket. As language models become more powerful, perhaps, they acquire System 2 abilities.

In enterprise settings, autonomous agents often take the form of personal assistants capable of conversing in natural language and executing actions that alter the state of the world (Kephart,

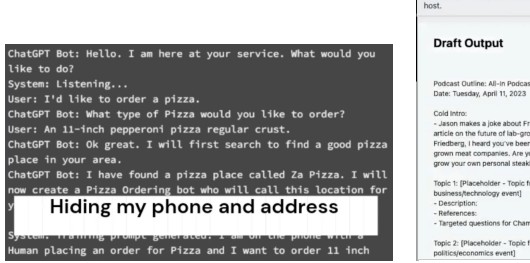

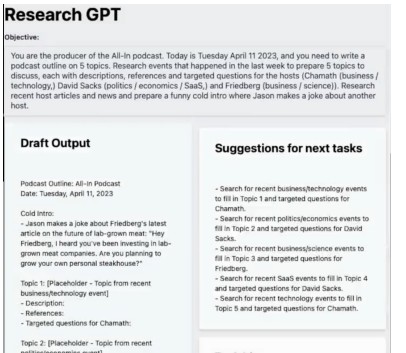

**Research GPT**

**Objective:**
You are the producer of the All-In podcast. Today is Tuesday April 11 2023, and you need to write a podcast outline on 5 topics. Research events that happened in the last week to prepare 5 topics to discuss, each with descriptions, references and targeted questions for the hosts (Chamath (business / technology,) David Sacks (politics / economics / SaaS,) and Friedberg (business / science)). Research recent host articles and news and prepare a funny cold intro where Jason makes a joke about another host.

**Draft Output**

Podcast Outline: All-in Podcast
Date: Tuesday, April 11, 2023

Cold Intro:
- Jason makes a joke about Friedberg's latest article on the future of lab-grown meat: "Hey Friedberg, I heard you've been investing in lab-grown meat companies. Are you planning to grow your own personal steakhouse?"

Topic 1: [Placeholder - Topic from recent business/technology event]
- Description:
- References:
- Targeted questions for Chamath.

Topic 2: [Placeholder - Topic from recent politics/economics event]

**Suggestions for next tasks**

- Search for recent business/technology events to fill in Topic 1 and targeted questions for Chamath.
- Search for recent politics/economics events to fill in Topic 2 and targeted questions for David Sacks.
- Search for recent business/science events to fill in Topic 3 and targeted questions for Friedberg.
- Search for recent SaaS events to fill in Topic 4 and targeted questions for David Sacks.
- Search for recent technology events to fill in Topic 5 and targeted questions for Chamath.

**Task List**

*Recipe: Lemon Berry Crostini with Goat Cheese*

*Ingredients:*
*-1 baguette, sliced into crostini rounds*
*-4 oz goat cheese, softened*
*-2 tbsp honey*
*-1 tbsp fresh lemon juice*
*-1/4 cup mixed berries (such as raspberries, blackberries, or blueberries)*
*-1/4 cup fresh mint, chopped*

*Instructions:*

*1. Preheat oven to 375°F. Place the baguette rounds onto a large baking sheet and bake until golden brown and crispy, about 6–8 minutes.*

*2. In a mixing bowl, combine the softened goat cheese with honey and lemon juice. Dip each crostini round into the goat cheese mixture.*

*3. Top the crostini with mixed berries and chopped mint.*

*Serve immediately and enjoy your delicious, unique Easter recipe!*

(a) Call a restaurant to order a pizza. (@AI-Growth-Startups, 2023)

(b) Perform research to prepare for a podcast. (@jamesbbaker4 , James Baker)

(c) Invent a receipt for an upcoming event. (Sharma, 2023)

Figure 2: Examples of autonomous agents that use LLMs to reason about a high level goal and compose tools such as search engines, phone services, and text to speech models, to achieve the goal.

2021; Rizk et al., 2020; Chakraborti et al., 2022). Among other functionality, these assistants carry out task-oriented dialog, further coupling language and reasoning. In addition to performing their functions accurately, enterprise-ready technology must be explainable and reliable. They must also adhere to strict regulations and governance, especially around concerns such as privacy, bias, and auditability. We notice, however, that beyond impressive demos, these LLM orchestrators are brittle, fail in unpredictable ways, and are not consistent which is not ideal for enterprise applications.

In this vision paper, we discuss the readiness of LLM-based autonomous agents for enterprise applications. First, we survey both anecdotal and academic works that have implemented LLM-based agents. Then, we show empirical evidence of LLMs as tool composers in an enterprise travel use case. We rely on both analyses to define a set of challenges that must be addressed to make LLMs resilient enough to use as an orchestrator for mission-critical enterprise use cases.

## 2   Anecdotal uses of LLM-based agents

ChatGPT created a lot of excitement around building LLM-based autonomous agents, where researchers and non-researchers alike were inspired to create demos, applications, and services using LLMs. In this section, we review anecdotal accounts of LLM-based autonomous agents and tool composers. They help stretch the imagination of what is possible, but do not perform experiments to stress test their solutions and understand the strengths and limitations of LLMs in this role.

### 2.1   Examples

An agent, inspired by Auto-GPT (Richards, 2023), was able to successfully order a pizza over the phone (@AI-Growth-Startups, 2023; @rogerhamilton , Roger James Hamilton); GPT-4 reasoned about the goal, asked clarifying questions such as the type of pizza, used tools to find a nearby restaurant and look up the phone number, made a phone call, and conversed with a real human (who did not realize they were conversing with a bot). Figure 2a shows some of the internal reasoning by this agent. While this was an impressive demo, even the author admitted that it took a few attempts to successfully place an order, illustrating the brittle nature of these agents. It was also a custom solution with a known task, and a fixed set of tools including speech-to-text, text-to-speech, and the Twilio API for phone calls. It is not clear how this would extend to support arbitrary goals and dynamically pluggable tools.

Another agent based on Auto-GPT and BabyAGI (Nakajima, 2023) concepts was given the task of preparing relevant topics for a podcast (@jamesbbaker4 , James Baker). Given the names and expertise of the hosts, it performed a series of web searches for current events related to the hosts' expertise, reasoned about whether the topics were relevant and covered the interests of all the hosts, and performed further searches to improve on the topics. Figure 2b depicts a partial output from this agent. It was restricted to web search as a tool and hence avoided issues with tools that may have undesirable side effects.

An agent was used to "browse the web to discover the next upcoming event and invent a unique

and original recipe that would suit it" (Sharma, 2023). It used GPT-4 to devise a five-step plan, including performing web searches to find an upcoming event, generating a relevant recipe, and writing the output to a file. Figure 2c shows the recipe this agent wrote to a file.

There are other examples of LLM-based agents performing sales prospecting (@ompemi, Omar Pera), autonomously carrying out tasks in a task list (Scott, 2023), and managing social media accounts (Sharma, 2023).

## 2.2 Limitations

A common observation on the LLM-based autonomous agent examples is that they tend to be one-off demos. Applying more scrutiny reveals issues around robustness, practicality, and trustworthiness. For example, an agent can get into loops where it continuously tries to decompose a task into smaller ones or generates new tasks that are duplicates of ones already in the task list or have been completed. This leads to agents performing work but not making progress towards the higher level goal (Xiao, 2023; Molony, 2023).

These agents can be brittle (e.g., sensitive to small changes in how the goals are expressed) and sometimes even producing different outputs for the same input (Brandt, 2023). The authors have also observed such brittleness when ChatGPT Plugins (OpenAI, 2023) was used to compose tools, with inconsistent behavior for the same or similar inputs (Patil et al., 2023). The agents are also sensitive to the LLM that is used, making it difficult to predict their behavior with different models or even newer versions of the same models. This lack of robustness is a challenge in enterprise applications.

Another observation is that these can be slow and expensive. They can make hundreds of calls to LLMs as they reason about the goals and make observations on the tool outputs. For example, even relatively simple goals can take 50 calls to GPT-4 costing about $14 (Xiao, 2023). This is not only expensive monetarily, but these iterative LLM calls can take several minutes or longer to accomplish a goal making them impractical for interactive apps.

Allowing agents to decide how to compose tools that have side-effects can be dangerous. Some use cases only compose tools such as web search with little harmful effects, while others utilize tools to send emails, write to a file system, or manipulate calendars. The agents and LLMs have little or no

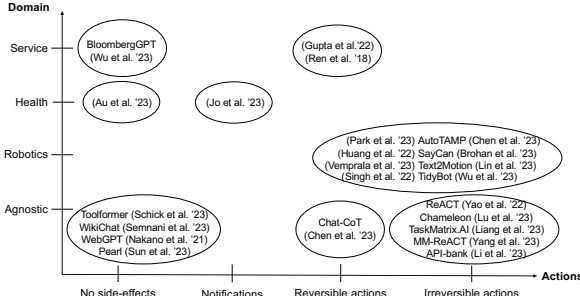

Figure 3: Spectrum of actions per domain

understanding of the risks of actions taken with a tool and may autonomously perform actions such as installing certificates as a super user (Molony, 2023). These risks are somewhat mitigated by having agents prompt the user at every step before acting, but this diminishes the value of these agents autonomously carrying out tedious work for users.

There are also studies that present the limitations of the reasoning abilities of LLMs (Borji, 2023). Since reasoning is such a critical part of how agents use LLMs, these limitations will have a direct effect on the performance of these agents.

These anecdotal observations of current limitations of LLM-based agents present roadblocks to adoption for enterprise use cases, where risk, cost, and robustness are critical issues. We formulate a set of research challenges in Section 5 to address these and other roadblocks.

## 3 A review of LLM-based agents

In the previous section, we provided evidence of the feasibility of LLM-based agents from non-academic sources such as blogs, Twitter posts, GitHub repositories, and demos. While these generated excitement around LLMs outside of the research community, this evidence does not allow us to assess the true capabilities and limitations of LLMs. Next, we survey published literature that more systematically analyzes LLMs in domains-agnostic and domain-specific settings. These papers consider a spectrum of actions from no side-effects to irreversible as illustrated in Figure 3.

**Domain-agnostic autonomous agents** Recent papers extend the use of LLMs beyond simple tasks such as summarization or entity extraction to handle complex reasoning and question answering. This includes augmenting LLMs with the ability to query the internet (Nakano et al., 2021; Semnani et al., 2023) or to use of different types of

tools (e.g. calculators) or access external information (e.g. Google, or Wikipedia search) (Schick et al., 2023; Chen et al., 2023c). When (Sun et al., 2023) leveraged LLMs to decompose complex tasks into sequence of actions that have no side effects, TaskMatrix.AI and ReACT present a vision that utilizes GPT-4 to connect any APIs to complete tasks (Liang et al., 2023; Yao et al., 2022; Yang et al., 2023b) (Lu et al., 2023) focused on plug and play when leveraging LLMs for tool composition. Finally, (Li et al., 2023) created a benchmark with 500+ APIs to evaluate the effectiveness of LLMs as tool composers.

As depicted in Figure 1, an evolution of these agents adds long term memory and task prioritization capabilities. An LLM decomposes high level goals to smaller tasks, a vector store is typically used for long term memory to recall outputs from long task lists, and an LLM is used to iteratively reprioritize existing tasks and create new tasks based on the progress towards the goal (Richards, 2023; Nakajima, 2023; Wang et al., 2023b).

**Robotic agents** Due to their emergent behaviors (Bommasani et al., 2021), LLMs have been used to make goal-driven decisions. These approaches mainly rely on LLMs to accomplish tasks in the physical or virtual world using Internet-of-Things devices and robots (Huang et al., 2022; Singh et al., 2022; Vemprala et al., 2023). (Huang et al., 2022) evaluate whether LLMs contain information necessary to accomplish goals without additional training or grounding. Followup efforts on grounding primitive actions include Text2Motion (Lin et al., 2023a), AutoTAMP (Chen et al., 2023b) and SayCan (Brohan et al., 2023) to guide LLM-based task and motion planning. TidyBot (Wu et al., 2023a) uses LLMs to infer generalized user preferences that are applicable to future interactions. LLM-based interactive agents mimicked human behavior in (Park et al., 2023).

**Other domains** Dialog systems with the goals of performing tasks such as finding restaurants or reserving hotels have been addressed in the NLP community (Gupta et al., 2022; Ren et al., 2018). For the banking sector, (Wu et al., 2023b) trained a finance specific LLM. In the medical domain, (Au Yeung et al., 2023) tested ChatGPT for clinical question-answering. (Jo et al., 2023) created a public health intervention system that can chat with patients and notify health care professionals

if intervention was necessary. In the area of online games, Voyager (Wang et al., 2023a) uses GPT-4 to build an agent to continuously explore a Minecraft world with the goal of making novel discoveries. It uses a catalog of skills or tools, and can also learn new skills.

# 4 Use case: chatbot for enterprise travel

The integration of APIs with LLMs (Liang et al., 2023) has raised the question of whether LLMs can act as task-oriented chatbots, particularly on complex tasks that require the execution of a sequence of APIs. Automating tasks in enterprise settings is particularly difficult since each company has custom processes in place and often little data to train or fine-tune an LLM to their peculiarities.

## 4.1 Travel example

Let's consider a travel example where LLM users may want to book a trip which includes making flight, hotel, and car rental arrangements. For enterprises, additional steps are necessary such as estimating travel costs and submitting a pre-approval with adequate justification (e.g., client event or conference travel), as shown in Table 1. Simply classifying these utterances into intent classes may not be enough as some of the information to distinguish between these clusters is not found in the natural language utterances provided by the users.

## 4.2 Methodology

Historically, process knowledge was embedded into task-oriented dialog systems by system developers. For example, a chatbot may ask the user (based on their underlying dialog tree) if their travel is a client event or a conference in cases where the process requires this information to determine the next step. By placing LLMs at the center of such dialog systems, how can we infuse such process knowledge into LLMs' decision making? In this case study, we focus on doing so through prompting but fine-tuning and possible pre-training models with appropriate data could also be considered.

The knowledge to make good decisions may come from multiple sources (Figure 4). For example, business process descriptions or standard operating procedures may include the necessary steps for business travel (e.g., submit a pre-approval first). Past process traces showing the execution of such processes can reveal information about best practices (e.g., most travel requests to conferences

| User Utterance | Cluster | Expected Sequence of APIs |
|---|---|---|
| BOOK MY TRIP FROM BOSTON TO NYC FOR MAY 2 | Business, Day trip | `Slot filling -> Car rental` |
| RESERVE MY TRIP TO PARIS FOR THE ACME WORKSHOP FROM MAR.1-4 | Business, Multi-day | `Travel Estimation -> Travel Pre-approval` |
| BOOK MY TRIP FROM BOS TO SFO FOR MAY 5-10 | Personal | `Slot filling -> Airline booking` |
| I'M ATTENDING EMNLP 2023 | Conference, Multi-day | `Publication database -> Travel estimate API -> Travel pre-approval` |

Table 1: Travel Example

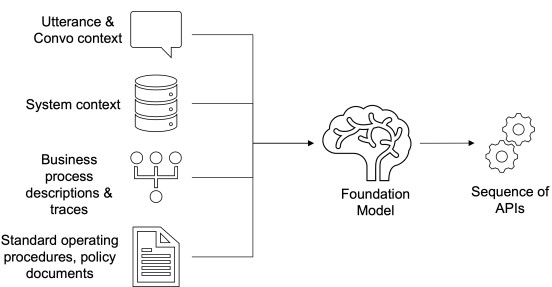

Figure 4: Sources of information to feed into an LLM

include listing papers accepted at the conference).

### 4.3 Experimental setup

**Model and prompt** To circumvent the need for a multi-modal foundation model, we represent all the sources of information in natural language, as in Figure 5. We experiment with a few variations of the prompt by paraphrasing some of its sections. Multiple versions of the prompts are fed into an instruction-tuned transformer model (FLAN-T5) (Longpre et al., 2023).

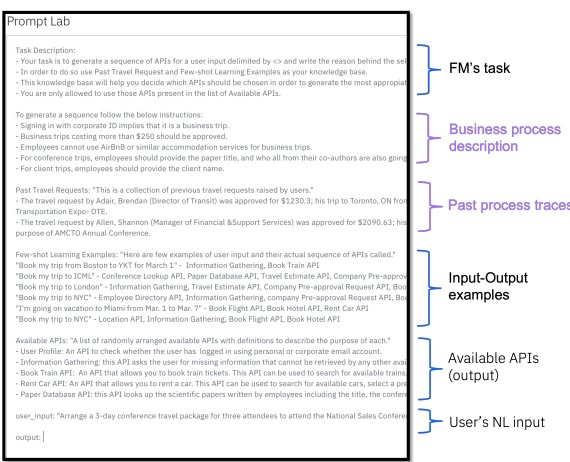

Figure 5: A prompt template for the travel use case

**Dataset** We created natural language utterances to sequence of API pairs (similar to Table 1) from two publicly available travel datasets (ap-

prox. 700 from Google employee travel (Emp, 2022) and 1200 from Frames (Fra, 2018). We manually labelled twelve phrases and then clustered the phrases to weakly label them.

**Evaluation metrics** To compare the various prompts, we adopted the BLEU metric (Papineni et al., 2002) and calculated precision, recall and F1-score as follows. Precision is the total number of common APIs between the ground truth sequence and the model's output divided by the total number of APIs in the model's output sequence. Recall is the total number of common APIs between the ground truth sequence and the model's output divided by the total number of APIs in the ground truth sequence. We calculated them at the set and sequence levels. Set level metrics compare the set of APIs in the ground truth and the output sequence, whereas sequence level metrics are based on the *longest common subsequence* between the ground truth and output sequence.

### 4.4 Experimental results

Tables 2-5 summarize our results. A baseline prompt (v1) does not include any process knowledge or past process traces (i.e., only the blue sections of Figure 5), whereas the remaining versions (v2-v5) include all sections in Figure 5 but paraphrased to express the same information in different ways (see Appendix for an example). We can see from the tables that including process information, not just users' natural language phrases and conversation context, outperformed the baseline prompt (in both set level and sequence level evaluations). Furthermore, the way that this information is included in the prompt matters (i.e., v4 and v5 perform better than other versions).

To evaluate LLMs' consistency, we prompted the model 30 times with a fixed input using a temperature of 0.5. Prompt v5 was used as the template and we compared the output of the model to the output when the temperature is set to 0. We did

| Prompt | Precision | Recall | F1-score | BLEU |
|--------|-----------|--------|----------|------|
| v1 | 0.32 | 0.39 | 0.35 | 0.38 |
| v2 | 0.38 | 0.48 | 0.43 | 0.19 |
| v3 | 0.61 | 0.76 | 0.68 | 0.55 |
| v4 | 0.61 | 0.78 | 0.68 | 0.57 |
| v5 | **0.65** | **0.79** | **0.71** | **0.62** |

Table 2: Comparing prompts on Google Employee Travel Dataset (averaged over instances). Precision, recall, and F-1 are calculated at set level.

| Prompt | Precision | Recall | F1-score | BLEU |
|--------|-----------|--------|----------|------|
| v1 | 0.54 | 0.55 | 0.54 | 0.42 |
| v2 | 0.57 | 0.55 | 0.56 | 0.40 |
| v3 | 0.55 | **0.61** | 0.58 | 0.40 |
| v4 | **0.68** | 0.60 | **0.64** | **0.56** |
| v5 | 0.63 | 0.55 | 0.59 | 0.52 |

Table 3: Comparing prompts on Frames Dataset (averaged over instances). Precision, recall, and F-1 are calculated at set level.

| Prompt | Precision | Recall | F1-score |
|--------|-----------|--------|----------|
| v1 | 0.28 | 0.34 | 0.31 |
| v2 | 0.33 | 0.44 | 0.38 |
| v3 | 0.56 | 0.69 | 0.62 |
| v4 | 0.55 | 0.70 | 0.62 |
| v5 | **0.61** | **0.73** | **0.69** |

Table 4: Comparing prompts on Google Employee Travel Dataset (averaged over instances). Precision, recall, and F-1 are calculated at sequence level.

| Prompt | Precision | Recall | F1-score |
|--------|-----------|--------|----------|
| v1 | 0.52 | 0.53 | 0.53 |
| v2 | 0.54 | 0.52 | 0.53 |
| v3 | 0.53 | **0.59** | 0.56 |
| v4 | **0.65** | 0.57 | **0.61** |
| v5 | 0.61 | 0.53 | 0.57 |

Table 5: Comparing prompts on Frames Dataset (averaged over instances). Precision, recall, and F-1 are calculated at sequence level.

| Phrase | BLEU | Precision | Recall | F1-score |
|--------|------|-----------|--------|----------|
| 1 | 0.56 | 0.46 | 1.00 | 0.63 |
| 2 | 0.86 | 0.92 | 0.89 | 0.91 |
| 3 | 0.95 | 0.93 | 0.97 | 0.95 |
| 4 | 0.48 | 0.51 | 1.00 | 0.68 |
| 5 | 0.70 | 0.75 | 0.74 | 0.74 |
| 6 | 0.74 | 0.90 | 0.78 | 0.84 |
| 7 | 0.81 | 0.84 | 0.88 | 0.86 |
| 8 | 0.97 | 0.97 | 1.00 | 0.99 |
| 9 | 0.68 | 0.72 | 0.72 | 0.72 |
| 10 | 0.77 | 0.96 | 0.82 | 0.89 |
| 11 | 0.71 | 0.82 | 0.93 | 0.87 |
| 12 | 0.91 | 0.85 | 0.97 | 0.91 |
| Average ± Standard Deviation | 0.76 ± 0.15 | 0.80± 0.17 | 0.89± 0.10 | 0.83± 0.11 |

Table 6: Repeatability on Manually Labelled Phrases

this for the 12 manually labeled phrases from the dataset, as shown in Table 6. Ideally, we want the metrics to be close to 1 to have a consistent LLM. In this case, we see that the results are not very repeatable with an average BLEU score of 0.76.

# 5 Research challenges

We now draw on the limitations observed from anecdotal applications of LLM-based agents (Section 2), experimental evaluation of an enterprise travel case study (Section 4), and our own experiences developing enterprise application platforms[1] to define a set of research challenges that need to be solved before LLM-based agents can be applied to mission-critical enterprise use cases.

## 5.1 Metrics and benchmarks

The evaluation of personal agents depends on the problem formulation. As a task-oriented dialog, it is commonly evaluated for accuracy of intent detection and accuracy of slot filling, i.e., how well the values of parameters detected from the natural language utterance match the ground truth API call. AST (abstract syntax tree) sub-tree matching is another metric to measure the correctness of API calls. The natural language utterance can be single-intent or multi-intent. This is the simplest form of personal agents and has existing benchmarks (Hemphill et al., 1990; Coucke et al., 2018; Qin et al., 2020; Patil et al., 2023). Most of these benchmarks contain a limited number (dozens) of APIs or tools with a limited number of slots per API and some of them are synthetically generated.

For a multi-turn dialog, it is also important to evaluate the number of turns required to achieve the goal. If the task requires executing a sequence of dependent tools, then the order of the tools becomes important. There are a few benchmarks available for this setting (Rastogi et al., 2020; Li et al., 2023; Budzianowski et al., 2018) with only one of them being a fully human annotated corpus (Budzianowski et al., 2018).

Furthermore, most of these datasets are simplified and do not represent enterprise scenarios. For example, (Li et al., 2023) has well-documented handcrafted APIs with a handful of parameters per API. Real tools on the other hand have many configuration options and are rarely well documented.

For a more complex setting of autonomous agents interacting with each other, the evaluation

---
[1]Citations omitted for double-blind review.

is challenging. Human evaluation is a critical part for building such agents (Park et al., 2023). If the tasks change the state of the world, then comparing the state after each action is a potential metric.

The development of representative benchmarks has accelerated the field of NLP. We believe this is an important area to focus on for enterprise personal agents, especially ones that factor in business process information and other modalities to fully capture the domain knowledge.

## 5.2 Data for fine-tuning

Most of the LLM training efforts on tasks in the natural language domain have large amounts and variety of relevant and (un)labeled training data that has been collected and open-sourced by the larger research community. In most enterprise settings, there is either a lack of sufficient labeled open-source real-world data to fine-tune a model due to their inherent proprietary nature, or lack of necessary infrastructure or training expertise that enables the fine tuning of such LLMs. One way to overcome this is by further advancements in prompt engineering (Wei et al., 2022) but also to provide a framework that efficiently guides developers to structure their prompts, and effectively prompt the model to generate more accurate results and enhance the overall performance without the need to fine-tune the models.

## 5.3 Composing tools

LLMs are not typically explicitly trained with the goal of composing tools and acting as the reasoning engine of an autonmous agent. Likewise, tools such as web search APIs, databases, and file system primitives aren't designed to called by a dynamic agent and LLM. We see some evidence of this impedance mismatch with ChatGPT Plugins requiring plugin developers to author manifest files for their existing tools, as well as extensive guidance on how to describe the tools, so they can be composed by ChatGPT (OpenAI, 2023).

More work is needed on improving the capabilities of LLMs to compose tools that take structured inputs and outputs, not just natural language. Furthermore, agents need an understanding of the risks and side-effects of performing actions with tools; a tool to search the web should be treated differently from one that performs financial transactions. This might require better interfaces or programming models to make these tools more consumable by agents and LLMs. A related aspect is these

efforts should improve the predictability and debuggability of these systems. As touched on in Section 2, current agents are brittle and hard to test.

## 5.4 Pluggability

An enterprise user needs to perform many tasks on a typical day. With an ever-evolving landscape of software tools, this set is large and anything but constant. Hence, personal assistants should allow for easy plugging in of new tools. Current prompt-based approaches such as ChatGPT function calling (Eleti et al., 2023) are limited by the context length of the model to add more APIs. Approaches such as (Schick et al., 2023) can potentially scale to a large number of APIs but new APIs cannot be added dynamically without re-training the model. Ensuring pluggability will also make it difficult to purely rely on fine-tuning models since the rate of adding and removing plugins and data scarcity could make fine-tuning prohibitive.

## 5.5 Reproducibility and reliability

The sensitive and business critical nature of enterprise tools and systems requires consistency and reliability from their tools and compositions. However, LLMs are capable of hallucinating predictions (Jiang et al., 2020), and could generate inconsistent and invalid tools and compositions, thereby having harmful consequences. Additionally, these models are brittle, where small variations in the prompt could result in different predictions, resulting in users' having very different experiences with the same model. Approaches like instruction-tuning (Ouyang et al., 2022) and chain-of-thought prompting (Wei et al., 2022) alleviate some of these problems by breaking down prompts into stages to improve prediction consistency. However, these approaches do not provide any guarantees on the reliability and reproducibility of the predictions.

An increasingly popular approach to enable reliable LLM predictions is constrained decoding (Hokamp and Liu, 2017), which enforces the model to only consider certain outputs for predictions by modifying their log-probabilities based on the given constraints. This would enable enterprise systems to prevent hallucinated outputs. Additionally, enterprises could also represent their policies as constraints to the model, to enforce compliance. However, as the number of constraints increases and in multi-modal settings, representing these constraints and policies in a format that the model can consume presents a significant challenge.

## 5.6 Confidence and failing gracefully

Data sources used to train LLMs are often restricted to positive knowledge and do not provide sufficient negative examples, including appropriate responses in case of failure or lack of knowledge, resulting in LLMs confidently producing incorrect answers (Chen et al., 2023a; Jiang et al., 2020). In enterprise settings, it is critical for LLM based systems to be able to recognize their limitations and fail gracefully. This is challenging to achieve, since the notion of uncertainty is dependent on the application domain, the knowledge sources used to train the model, and an evaluation of the model's response. While there has been some initial efforts to develop methods to measure the confidence or uncertainty of black-box models (Lin et al., 2023b; Kuhn et al., 2023), they do not translate to many enterprise use-cases such as those described in this paper, necessitating specialized approaches.

## 5.7 Error handling and failure semantics

As indicated previously in Section 2.2, LLM agents can get into loops trying to decompose a task or perform actions that can have lasting side effects on their environment. Handling such situations will be necessary to provide a reliable and consistent output in an enterprise setting. Furthermore, as LLM's access to tools increases, it becomes critical for them to overcome errors such as page not found, non-responsive server, or unauthorized access, that can get them stuck and hinder their progress.

## 5.8 Multi-modality

Many enterprise applications have decades worth of information saved in various modalities such as spreadsheets, scanned documents in image format, unstructured emails, business process diagrams and others. One way to address this diversity is to bring everything into the language space (per section 4). Prior work has shown that better performance can be achieved by learning a unified representation across modalities (Bao et al., 2022) as opposed to treating these modalities independently (Jia et al., 2021) or converting into one modality where information will be inevitably lost (Xiang et al., 2023). Creating a new class of foundation models that include LLMs along with models for enterprise specific modalities like business processes is worth investigating (Rizk et al., 2022).

## 5.9 Generalizability and scalability

Most popular LLMs have a large number of parameters, making them prohibitively expensive to fine-tune for different tasks. Given the varied objectives of enterprise use-cases, ensuring the generalizability of models to new tasks and domains, and unforeseen tools is essential. Approaches like few-shot prompting and prompt tuning have become more popular (Lester et al., 2021; Gu et al., 2022), leveraging specific and well-crafted examples to improve model generalizability. The growing size also inherently increases inference time and infrastructure costs, and enterprise services that depend on these models often have latency constraints. Addressing the scalability and costs of LLM-driven enterprise applications is another significant challenge that requires attention.

## 5.10 General purpose or specialized models

In Figure 1, a single LLM is used for multiple tasks including task reasoning and composing tools. There is no reason to presuppose that a single model is well-suited to every step in that flow. Thus, thought should be given to whether specialized models for each task perform better and how to chain them to complete the overall objective.

## 6 Final thoughts and next steps

Enterprises are on a continuous quest to optimize repetitive and tedious work. Advancements in LLMs have broadened imaginations on what knowledge work can be automated, and LLM-based autonomous agents can be the vehicle to deliver incredible productivity gains.

Attempts in the community to build agents using even state-of-the-art LLMs, as well as our case study using LLMs for an enterprise travel use case, reveal how inadequate these solutions are for mission-critical enterprise use cases. We outline several research challenges that the NLP and other research communities can investigate.

While we describe individual challenges, such as robustness, multi-modality, and tool composition, the community must take a holistic view of the problem, addressing not just LLM advances, but the entire software lifecycle including tool authoring, and debugging. Defining clear use cases and benchmarks will be important steps.

## Limitations

On the survey and challenges contributions of this work: our coverage of the literature is incomplete due to the extraordinary fast pace and sheer volume of work posted on arxiv, blogs, etc. Furthermore, due to this rapid pace, we do cite a large number of non-peer-reviewed work. Our discussion of existing challenges and how to resolve them is also colored by our experience in industry and may not include some more theoretical challenges that the community should also solve in conjunction with the practical challenges.

On the experimental contributions of this work: our datasets are small in size compared to what may be commonly used, their labeling is noisy due to the pseudo-labeling approach we adopted and may not be very realistic due to the additional processing we performed to get them to the format we required. Furthermore, the metrics we calculated do not measure all characteristics that we may want to evaluate. Also, since we only used one ground truth to calculate the metrics, this may result in less accurate values (e.g., BLEU becomes more accurate as more references are used). Finally, the experimental analysis is not comprehensive, missing some ablation studies and other experiments that could help answer additional questions on the performance on LLMs on task-oriented dialog tasks.

## Ethics statement

Our work discusses how to enable LLMs to perform actions (or compose tools) whose purpose is to change the state of the real world. Given the emergent property of LLMs that may result in unpredictable behavior, allowing these LLMs to alter the state of the world by performing these actions could lead to irrevocable changes that could have negative impact (e.g., autonomous agents capable of stealing money from people's banks).

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

# Appendix

## A  Prompts

Check supplemental material for complete prompts. Figure 6 shows an example of the difference in how the rules are phrased between v2 and v4.

Prompt v2 - Rules

```
7   Follow the below instructions:
8   – Signing in with corporate ID implies that it is a business trip.
9   – Business trips costing more than $250 should be approved.
10  – Employees cannot use AirBnB or similar accommodation services for
11  – For conference trips, employees should provide the paper title, a
12  – For client trips, employees should provide the client name.
```

Prompt v4 - Rules

```
8   If <travel_type> is business, follow the below rules:
9   – Business travels costing more than $250 should be approved using Trave
10  – If it's a conference trip, use Paper Database API & Employee Directory
11  – For client trips, use Client Database API to verify the client.
12
13  If <travel_type> is personal, follow the below rules:
14  – Do not use Travel Estimate API or Company Pre-approval Request API in
```

Figure 6: Rephrasing of rules in prompt v2 and v4

## B  Datasets

**Employee Travel/Golden dataset**   We have used freely available Employee Travel 2022 dataset from Google Dataset Search. This dataset contains information about the employee travel expenses for the year 2022. Details are provided on the employee (name, title, department), the travel (dates, location, purpose) and the cost (expenses, recoveries). Count of rows = 178.

For the purpose of generating NL utterances, we used four columns (termed as entities) - dates (start and end), location and purpose of travel. Using these entities, we divide our NL utterance generation into four categories:

- random_sample_4: with all the four entities present (start date, end date, purpose of travel and location of travel)

- random_sample_3: with any three random entities present

- random_sample_2: with any two random entities present

- random_sample_1: with any one random entity present

Using OpenAI's text-davinci-003 model, we engineered a prompt after preprocessing the dataset, to generate five distinct user utterances with the help of the entities, in first person. The same process was repeated for all the defined categories above. This approach generated an overall count of 3560 utterances from the initial set of 178 rows.

**Past Travel Requests**   We created a template, using seven entities, to craft travel requests made by employees in the past. These past travel requests are a part of our prompt.

Template: The travel request by <name> (<title>) was approved for <amount>; his trip to <destination> from <date> to <date> was for the purpose of <purpose>

Example: 'The travel request by Adair, Brendan (Director of Transit) was approved for $1230.3; his trip to Toronto, ON from 2022-07-17 to 2022-07-20 was for the purpose of Ontario Transportation Expo- OTE'

**Frames Dataset** Frames is a dialogues dataset which was collected in a Wizard-of-Oz fashion by (El Asri et al., 2017). Two humans talked to each other via a chat interface. One was playing the role of the user and the other one was playing the role of the conversational agent called wizard. The wizards had access to a database of 250+ packages, each composed of a hotel and round-trip flights.

Frames is composed of 1369 human-human dialogues with an average of 15 turns per dialogue. This corpus contains goal-oriented dialogues between users who were given some constraints to book a trip and assistants who search a database to find appropriate trips.

The Frames dialogues are in JSON format. Each dialogue has five main fields: user_id, wizard_id, id, userSurveyRating and turns. For our purpose, we extracted the first occurrence from every user-wizard dialogue or id. On postprocessing, we were able to create a dataset of about 1200 user utterances. The postprocessing was done using HDB-SCAN to group similar sentences into clusters and then discarding those clusters which contained only greetings (such as Hi, Hello there) without any request for travel booking.

## C Metrics

We will calculate our set level and sequence level metrics for an example utterance. Consider the following utterance with the corresponding ground truth and model output sequence.
`utterance`: *I'm planning a ski trip to Banff, AB in February. Can you help me plan it out?*
`output_sequence`:{Information Gathering API, Book Hotel API, Book Flight API, Book Train API}
`gound_truth`:{Information Gathering API, Book Flight API, Book Hotel API, Rent Car API}
For set level metrics we will look at the intersection of the *set* of APIs in `output_sequence` and `ground_truth`. We have:
set(`output_sequence`) $\bigcap$ set(`ground_truth`) =
{Information Gathering API, Book Hotel API, Book Flight API}
Then, set level precision is:

$$\frac{|set(\texttt{output\_sequence}) \bigcap set(\texttt{ground\_truth})|}{|\texttt{output\_sequence}|} = \frac{3}{4},$$

and set level recall is:

$$\frac{|set(\texttt{output\_sequence}) \bigcap set(\texttt{ground\_truth})|}{|\texttt{ground\_truth}|} = \frac{3}{5}.$$

For sequence level metrics, we will look at the longest common subsequence (LCS) between the *sequence* of APIs in `output_sequence` and `ground_truth`. :
LCS(`output_sequence`, `ground_truth`) = {Information Gathering API, Book Hotel API}.
Then, the sequence level precision is:

$$\frac{|LCS(\texttt{output\_sequence}, \texttt{ground\_truth})|}{|\texttt{output\_sequence}|} = \frac{2}{4},$$

and the sequence level recall is:

$$\frac{|LCS(\texttt{output\_sequence}, \texttt{ground\_truth})|}{|\texttt{ground\_truth}|} = \frac{2}{5}.$$

Observe that another common subsequence is {Information Gathering API, Book Flight API} which is also of length 2. If we use this instead, we will get the same sequence level precision and recall.