# OpenReview forum: "Towards large language model-based personal agents in the enterprise: Current trends and open problems"
_EMNLP/2023/Conference — EMNLP 2023 Findings_

### Official Review · Reviewer_dPLF · 2023-08-03

**Soundness:** 4

**Excitement:**

4: Strong: This paper deepens the understanding of some phenomenon or lowers the barriers to an existing research direction.

**Paper Topic And Main Contributions:**

This vision paper discusses the current state of LLMs as autonomus agents for enterprise appllications.  The authors survey both anecdotal and academic work that have implemented LLM-based to stuate the discussion. Then, empirical evidence on the use of LLMs as tool composers in an enterprise travel use case is provided. Finally, the insight obtain from those analyses is used to define a set of challenges and recommendations to enable LLMs to be used in mission-critical orchestration enterprise use cases.

The paper does a good job showing the current state of LLMs as agents for enterprise applications, including empirical evidence. Also in providing the necessary insight to motivate the final list of recommendations, which IMO look reasonable and point at interesting future work.

**Reasons To Accept:**

+ Visionary and grounded in enough evidence
+ Lays out a number of interesting reseach direction focused on solving specific problems
+ A joy to read

**Reasons To Reject:**

- The paper may benefit from complementing the travel example with other cases in other domains.

**Reproducibility:**

4: Could mostly reproduce the results, but there may be some variation because of sample variance or minor variations in their interpretation of the protocol or method.

**Reviewer Confidence:**

4: Quite sure. I tried to check the important points carefully. It's unlikely, though conceivable, that I missed something that should affect my ratings.

---

> ### Author Rebuttal · Authors · 2023-08-25
>
> We thank you for the review and encouraging feedback.
>
> Regarding other domains beyond the travel example in Section 4 of the paper, we do survey other application use cases in Sections 2 and 3. The travel example serves to complement these surveyed domains with one that we can more deeply evaluate and further inform the research challenges listed in Section 5.

---

### Official Review · Reviewer_Jf2r · 2023-08-04

**Soundness:** 4

**Excitement:**

4: Strong: This paper deepens the understanding of some phenomenon or lowers the barriers to an existing research direction.

**Paper Topic And Main Contributions:**

The paper presents examples of LLM-based autonomous agents that reason and compose tools, highlights cases where they fail, surveys recent efforts in this space, and lays out the research challenges to make these solutions viable for enterprises.

**Questions For The Authors:**

1-	In line #072 you say: “In addition to performing their functions accurately, enterprise-ready technology must be explainable and reliable”, but no discussion on explanability is done. Why?

**Reasons To Accept:**

1-	The paper presents the SOTA on large language model-based personal agents in the enterprise;
2-	Since “LLM-based autonomous agents” is a new area of interest, the paper arrives at a good moment;
3-	The paper raises good points of attention in the Research Challenges section.


**Reasons To Reject:**

1-	The paper does not present a novelty but contributes to disseminating important facts on LLM in autonomous agents research.

**Reproducibility:**

3: Could reproduce the results with some difficulty. The settings of parameters are underspecified or subjectively determined; the training/evaluation data are not widely available.

**Reviewer Confidence:**

4: Quite sure. I tried to check the important points carefully. It's unlikely, though conceivable, that I missed something that should affect my ratings.

---

> ### Author Rebuttal · Authors · 2023-08-25
>
> We acknowledge that the paper does not present a novel solution. Please keep in mind that this is a vision paper “with the goal of stimulating discussion around Large Language Models and the Future of NLP” (as stated in the CFP for the LLM theme track). In any case, we think there is some non-obvious material in Section 5 where we define a set of 10 research topics that can provide guidance for the research community, including highlighting existing research to build on. We also think the paper brings attention to the community about how LLMs are being used as autonomous agents in practice (Section 2), in the literature (Section 3), and with an empirical use case (Section 4) that can be leveraged by the research community to improve LLM performance (especially in applications that are relevant for enterprises).
>
> Regarding the reviewer’s question about explainability and reliability, there is well-known literature about the pitfalls of relying on opaque AI models for use cases that have consequential effects on people [1], and the different dimensions of trust that users in high-stakes use cases expect [2]. We can include some of these references in the final version of the paper. Note, however, that several of the research areas identified in the paper are related to explainability and reliability, including Sections 5.5 (Reproducibility and reliability), 5.6 (Confidence and failing gracefully), and 5.7 (Error handling and failure semantics). Each of these subsections include discussions and relevant references.
>
> [1] Julia Angwin, Jeff Larson, Surya Mattu and Lauren Kirchner. “Machine Bias: There’s software used across the country to predict future criminals. And it’s biased against blacks.” ProPublica (2016). https://www.propublica.org/article/machine-bias-risk-assessments-in-criminal-sentencing
> [2] Varshney, Kush R. "Trustworthy machine learning." Chappaqua, NY (2021). http://trustworthymachinelearning.com

---

### Official Review · Reviewer_AqLF · 2023-08-06

**Soundness:** 4

**Excitement:**

4: Strong: This paper deepens the understanding of some phenomenon or lowers the barriers to an existing research direction.

**Missing References:**

As far as I could tell, the presented references are enough.


**Paper Topic And Main Contributions:**

This paper draws a review of the performance of LLM to diverse tasks that were approached through specific AI solutions trying to indicating the boundaries of effectiveness of the generic LLM approach compared to the specific solutions.

**Questions For The Authors:**

In your classification of agents you have divided it into domain-agnostic autonomous agents, robotic agents, and others. What other kind of classification could be done to be more problem oriented than solution oriented?

**Reasons To Accept:**

The topic is extremely relevant as several other submissions to EMNLP precisely use the generic LLM solution to problems traditionally solved with specific methods. Even though the paper analyzes anecdotal cases to draw its conclusions, the analysis is systematic and interesting insights can be perceived by the experiments.

**Reasons To Reject:**

The work is a preliminary analysis about the boundaries of effectiveness of the generic LLM approach, as much of the current applications is not discussed and probably a taxonomy of the problems would benefit the analysis. Yet the paper has the merit to call the reader the attention to the wide spread use of LLM without actually knowing how effective it might be to specific cases.

**Reproducibility:**

3: Could reproduce the results with some difficulty. The settings of parameters are underspecified or subjectively determined; the training/evaluation data are not widely available.

**Reviewer Confidence:**

3: Pretty sure, but there's a chance I missed something. Although I have a good feel for this area in general, I did not carefully check the paper's details, e.g., the math, experimental design, or novelty.

**Typos Grammar Style And Presentation Improvements:**

As far as I could tell, there were no typos or grammar issues.

---

> ### Author Rebuttal · Authors · 2023-08-25
>
> We agree that this is a preliminary analysis of the space of LLM-based autonomuous agents. Please keep in mind that much of the work in this space is only a few months old, and we submitted this vision paper “with the goal of stimulating discussion around Large Language Models and the Future of NLP” (as stated in the CFP for the LLM theme track).
>
> In terms of “current applications” in this space, we tried to survey both anecdotal accounts (Section 2) and the literature (Section 3). If the reviewer has important missing applications in mind, we would be happy to include them.
>
> About the “taxonomy of the problems”, we have dedicated Section 5 to research challenges, and have outlined 10 sets of problems that could each be future research topics. This is not explicitly  a “taxonomy” in terms of a structured hierarchy of problems, but we feel it still serves to classify the problem areas and provide some guidance to the research community. We will consider the reviewer’s advice for a more comprehensive problem taxonomy in a followup paper as we better understand the developments in this space.
>
> Regarding the reviewer’s question on classification schemes, note that we structured Section 3 into “domain-agnostic autonomous agents, robotic agents, and others” mainly because that reflects the balance of literature in the community. Figure 3 outlines another classification across two dimensions: the problem domain and the type of actions (from those with no side-effects such as searching the web, to those with irreversible side-effects). The classification by action type is important since it influences the solution design (e.g., whether the agent can explore and backtrack) and limits the use cases that a solution can be applied to (e.g., an agent that assumes actions have no side-effects may not be suitable for critical health care use cases). Another classification can be: information retrieval agents vs. personal assistant agents vs. task-oriented agents (Fig. 3 x-axis is a more specific view of this classification where info retrieval agents can be viewed as agents that take actions without any side effects or reversible actions). The reason we chose this specific classification in Fig. 3 was to lead to a conversation about challenges that may hinder LLM adoption in enterprises (e.g., many enterprises may care about task-oriented chatbot problems that may take irreversible actions and LLMs are error prone and hallucination prone which is problematic when their mistakes cannot be undone). We can make the link between the problem oriented classification and solution oriented classification more explicit in the final version.

---

### Meta-Review · Area_Chair_qzdb · 2023-09-19

**Recommendation:** 3

**Metareview:**

Overall, this paper addresses the topic of LMs powering agents with clarity, providing a structured framework* for how to reason about it.

As the reviewers all note, the paper is well-written and provides good conceptual clarity. I am perhaps a bit more critical than the reviewers here on how much the paper has sharpened our view on the agent paradigm; I would encourage the authors to be a bit more incisive about how things should evolve and so forth (e.g. most of the guidance in the paper is not terribly sharp/fairly restrained). With that said, this paper is a good contribution to our field and hopefully will help the growth of this subarea!

* Remark: I am not sure I would call the paper a vision paper/it seems more like a framework (whereas I would expect a vision paper to make more speculation about the future/take a stronger stand on how the future should evolve), but that's fine/a minor nit.

---

### Decision · Program_Chairs · 2023-10-07

**Decision:**

Accept-Findings

**Comment:**

Overall, this paper addresses the topic of LMs powering agents with clarity, providing a structured framework* for how to reason about it.

As the reviewers all note, the paper is well-written and provides good conceptual clarity. I am perhaps a bit more critical than the reviewers here on how much the paper has sharpened our view on the agent paradigm; I would encourage the authors to be a bit more incisive about how things should evolve and so forth (e.g. most of the guidance in the paper is not terribly sharp/fairly restrained). With that said, this paper is a good contribution to our field and hopefully will help the growth of this subarea!

* Remark: I am not sure I would call the paper a vision paper/it seems more like a framework (whereas I would expect a vision paper to make more speculation about the future/take a stronger stand on how the future should evolve), but that's fine/a minor nit.